# AntiGan: An Epinutraceutical Bioproduct with Antitumor Properties in Cultured Cell Lines

**DOI:** 10.3390/life12010097

**Published:** 2022-01-10

**Authors:** Olaia Martínez-Iglesias, Ivan Carrera, Vinogran Naidoo, Ramón Cacabelos

**Affiliations:** EuroEspes Biomedical Research Center, International Center of Neuroscience and Genomic Medicine, 15165 Bergondo, Corunna, Spain; biotecnologiasalud@ebiotec.com (I.C.); neurociencias@euroespes.com (V.N.); rcacabelos@euroespes.com (R.C.)

**Keywords:** cancer, nutraceutical, AntiGan, apoptosis, epigenetics, DNMT, sirtuin

## Abstract

Novel and effective chemotherapeutic agents are needed to improve cancer treatment. Epidrugs are currently used for cancer therapy but also exhibit toxicity. Targeting the epigenetic apparatus with bioproducts may aid cancer prevention and treatment. To determine whether the lipoprotein marine extract AntiGan shows epigenetic and antitumor effects, cultured HepG2 (hepatocellular carcinoma) and HCT116 (colorectal carcinoma) cell lines were treated with AntiGan (10, 50, 100, and to 500 µg/mL) for 24 h, 48 h, and 72 h. AntiGan (10 µg/mL) reduced cell viability after 48 h and increased Bax expression; AntiGan (10 and 50 µg/mL) increased caspase-3 immunoreactivity in HepG2 and HCT116 cells. AntiGan (10 and 50 µg/mL) attenuated COX-2 and IL-17 expression in both cell lines. AntiGan (10 µg/mL) increased 5mC levels in both cell types and reduced DNMT1 and DNMT3a expression in these cells. AntiGan (10 and 50 µg/mL) promoted DNMT3a immunoreactivity and reduced SIRT1 mRNA expression in both cell types. In HCT116 cells treated with AntiGan (10 µg/mL), SIRT1 immunoreactivity localized to nuclei and the cytoplasm; AntiGan (50 µg/mL) increased cytoplasmic SIRT1 localization in HCT116 cells. AntiGan is a novel antitumoral bioproduct with epigenetic properties (epinutraceutical) for treating liver and colorectal cancer.

## 1. Introduction

Cancer, a major cause of mortality worldwide, is characterized by unrestrained cell growth and the acquired ability to spread to other organs [1]. Cancer and malignant neoplasms were the leading cause of death in the 45- to 64-year-old age group in the United States in 2020; about 1.8 million individuals were diagnosed with cancer, and 600,000 people died from the disease [2]. Colorectal cancer (CRC) accounts for 10% of global cancer incidence and 9.4% of cancer-related deaths, just below lung cancer, which accounts for 18% of fatalities [3]. CRC, the third most frequent malignancy and the second most lethal cancer subtype, produces approximately 1.9 million new cases annually [3]. According to projections of aging, population expansion, and human progress, the number of new cases of CRC, globally, is expected to reach 3.2 million by 2040 [3]. East Asia and Africa are regions with the highest burden of hepatocellular carcinoma (HCC). After lung, colorectal, and stomach cancer, liver cancer is the sixth most often diagnosed cancer and the fourth greatest cause of cancer-related mortality globally [4].

Dysregulated cellular metabolism is a hallmark of cancer and occurs because of accumulated genetic mutations, epigenetic changes, and environmental factors. Epigenetics, the study of reversible and heritable alterations in gene expression and cellular function caused by chromatin-based mechanisms without changes to the original DNA sequence, is influenced by environmental factors [5]. Aberrant alterations to the epigenetic apparatus, therefore, promote inappropriate onset of genetic expressions and tumorigenesis [6]. Classic epigenetic mechanisms include DNA methylation, chromatin remodeling/histone modifications, and microRNA (miRNA) regulation [5]. Deficient epigenetic alterations, such as promoter hypermethylation of tumor suppressor genes and abnormal post-translational histone modifications caused by acetylation/methylation dysregulation and changes in miRNA expression, are associated with various types of cancers [7].

DNA methylation remains the epigenetic hallmark most investigated. DNA methylation is a reversible mechanism in which DNA methyltransferases (DNMTs) catalyze the transfer of methyl groups from the cofactor S-adenosyl methionine to the C5 position of cytosines in CpG dinucleotides; these cytosines are converted to 5-methylcytosines (5mC), affecting DNA stability and accessibility, thus regulating gene expression [8,9]. The DNMT family comprises the canonical enzymes DNMT1, DNMT3a, and DNMT3b. DNMT1 preserves methylation marks on newly synthesized DNA strands and maintains methylation pattern inheritance; de novo DNMTs include DNMT3a and DNMT3b, which generate methylation patterns [9]. The biological significance of cancer-associated DNA hypomethylation in the human genome is not well understood [10]. Global genomic hypomethylation occurs in various types of cancers, including hepatocellular and colorectal cancer [7,11,12,13]. Repetitive elements in heterochromatin, such as satellite repeats and retrotransposons, are hypomethylated in various cancer types, affecting chromatin structure and genomic stability [14]. Hypomethylation of interspersed repeat (e.g., retrotransposon long interspersed element-1, LINE-1) is found in human hepatocellular carcinoma [15].

Histone acetylation involves the transfer of an acetyl group to a lysine residue at the N-terminus of histones; this reduces the positive charge on histones, weakening their interaction with negatively charged DNA [16]. Histone acetylation promotes gene transcription by facilitating the binding of transcription factors and related enzymatic complexes to DNA. However, histone deacetylases produce the opposite effect and inhibit gene expression [16]. Sirtuins were discovered as transcription repressors in yeast but are now known to exist in bacteria and eukaryotes, including mammals [17]. The sirtuin family of proteins in humans comprises seven class III histone deacetylases (SIRT1–SIRT7). Sirtuins differ in their enzymatic activities, subcellular localizations, and physiological functions. All of them, however, are NAD-dependent protein deacetylases/ADP ribosyltransferases that play important roles in chromatin structure, cell cycle regulation, cell differentiation, cell stress response, metabolism, and aging processes [17]. SIRT1, the most extensively studied mammalian sirtuin [18], regulates neuronal differentiation, tumor progression, apoptosis, DNA and chromosomal structure stability, gene expression, and cell cycle progression [18].

Cancers are associated with aberrant epigenetic modifications such as alterations in DNA methylation or abnormal post-translational histone modifications by dysregulation of acetylation and/or methylation [7]. As epigenetic modifiers are sensitive to extrinsic factors and exhibit reversible mechanistics, they are emerging as promising targets in a variety of pathologies, including cancer. Several epidrugs, compounds that target enzymes with epigenetic activity or the epigenome, have been used in clinical trials for cancer therapy [6,19]. However, significant toxic effects have been reported with those agents [19]. Nutraceuticals also function as epidrugs [20,21,22,23], and several natural bioproducts such as quercetin, resveratrol, curcumin, genistein, and catechins, exhibit potent antitumoral effects by reverting epigenetic alterations associated with oncogene activation or tumor suppressor gene inactivation [24]. These compounds modulate the epigenome via chromatin remodeling mechanisms [25,26]. Nutraceuticals may, therefore, restore carcinogenesis-induced epigenetic alterations and represent an alternative therapeutic option for cancer treatment to help improve cancer outcomes [27,28,29].

AntiGan is a nutraceutical belonging to the lipofishin family, which has been developed through non-denaturing biotechnological procedures from the epidermis and esophagus of the sea eel *Conger conger* that shows immune-chemical fortification properties [30]. E-Congerine-10423 extract is the structural base of AntiGan and contains all the biological properties of the original species. The nutritional composition of E-Congerine-10423 highlights the high protein content (80%) and the large amount of unsaturated fatty acids (65% of the total fat). The main monounsaturated fatty acid in AntiGan is oleic acid, and the polyunsaturated fatty acids are represented mostly by EPA, DHA, and linoleic acid, and the most abundant saturated fatty acid is palmitic acid [31]. E-Congerine-10423 also has vitamins A and D in large quantities, and B vitamins, principally B1 and B3, and minerals, such as phosphorus, potassium, and magnesium. Congerin I and II are two proteins isolated from the skin mucus of conger eel (*Conger myriaster*), which belong to the galectin family of lectin proteins [20,32]. Galectins participate in many physiological processes such as the development, differentiation, morphogenesis, immunity, apoptosis, and metastasis of malignant cells. The existence of congerin proteins in the mucus skin of *Conger*
*conger* may be responsible for the immunomodulatory, cytotoxic, and chemopreventive effects of AntiGan [20,31,32,33]. AntiGan is cytotoxic to colorectal adenocarcinoma cells in culture and against colon cancer-related inflammation in vivo [34]. In humans, AntiGan reduces tumor marker levels in healthy subjects; this response is pronounced in patients with different types of cancer [31,34,35]. Those studies, however, did not test the potential for AntiGan as a nutraceutical product with epigenetic properties (epinutraceutical). Other nutraceuticals such as E-PodoFavalin-15999 (AtreMorine), a novel compound extracted from *Vicia fabia L*. and rich in 3,4-dihydrophenyl-l-alanine (L-DOPA), also regulate DNA methylation, but in neurodegenerative disorders [36,37,38]. In the present study, we investigated, in vitro, the effect of AntiGan against hepatocellular carcinoma, its cytotoxic action against colorectal cancer, and its potential as an epidrug. Given that AntiGan exhibits antitumor properties, the aim of this study was to determine whether AntiGan acts as an epinutraceutical by regulating the epigenetic machinery that drives cancer development and progression.

## 2. Materials and Methods

### 2.1. Cell Lines

Hepatocarcinoma HepG2 and colorectal adenocarcinoma HCT116 cell lines were maintained in Dulbecco’s modified Eagle’s medium (DMEM; Gibco, New York, NY, USA) supplemented with 1% penicillin/streptomycin (Gibco) and 10% heat-inactivated fetal bovine serum (Gibco). Cells were incubated at 37 °C in a humidified incubator with 5% CO_2_. HepG2 and HCT116 cell lines were kindly provided by Dr. A. Aranda (Instituto de Investigaciones Biomédicas “Alberto Sols”, Consejo Superior de Investigaciones Científicas, Universidad Autónoma de Madrid, Spain) and Dr. A. Figueroa (Epithelial Plasticity and Metastasis Group, Instituto de Investigación Biomédica de A Coruña, A Coruña, Spain), respectively.

### 2.2. Preparation of AntiGan (E-Congerine-10423) and Treatments

AntiGan was developed through biotechnological non-denaturing processes from the muscular structures and skin of the sea eel *Conger conger*. AntiGan contains essential amino acids, natural mono- and polyunsaturated fatty acids (mainly of the omega 3 type), vitamins, and minerals (Table 1). A stock solution (20 mg/mL) of lyophilized AntiGan was sonicated in sterile filtered 0.9% NaCl and centrifuged at 300× *g* for 3 min. The supernatant was then collected and used for all cell culture experiments.

For immunofluorescence assays, 1.25 × 10^5^ cells were grown on poly-L-lysine-treated glass coverslips in 24-well plates for 24 h at 37 °C. For global DNA methylation and gene expression studies, 4.5 × 10^5^ cells were grown for 24 h in 6-well plates at 37 °C. Cells were then exposed to 10 µg/mL and 50 µg/mL AntiGan for 48 h.

### 2.3. Immunofluorescence

Cells were washed twice with phosphate-buffered saline (PBS, pH 7.4), fixed in 4% paraformaldehyde for 15 min, and permeabilized with 0.1% Triton in PBS for 5 min. The cells were then washed 3 × 10 min with PBS and incubated in blocking buffer (2% bovine serum albumin in PBS) for 1 h at room temperature. This was followed by an overnight incubation at 4 °C with primary antibodies (Table 2), diluted in blocking buffer.

Negative primary controls included cells treated with blocking solution but without the addition of primary antibodies. The cultured cells were washed 3 × 10 min with PBS and incubated with Alexa-488- or Alexa-555-conjugated secondary antibodies (Thermo Fisher Scientific, Waltham, MA, USA), diluted 1:500 in blocking buffer for 2 h at RT. The cells were then washed three times with PBS and counterstained with 4′,6-diamidino-2-phenylindole (DAPI) for 15 min. Coverslips were mounted onto microscope slides using Vectashield antifade mounting medium (Vector Labs, Burlingame, CA, USA). Immunostained images were captured with a Leica DM6 B upright microscope (Leica Microsystems, Buffalo Grove, IL, USA) and Leica Application Suite X (LAS X) software. For each well, area/pixel analysis software (Pixcavator 4) was used to quantify the number of pixels inside the outer boundary of each cell body; this aided quantification of the density of immunofluorescence cell markers relative to background.

### 2.4. Cell Viability Assay

Cell viability was determined using PrestoBlue Cell Viability assay (Thermo Fisher). Cells (1 × 10^4^) were incubated in 96-well plates with AntiGan (10, 50, 100, and 500 µg/mL) for 24 h, 48 h, or 72 h. PrestoBlue reagent (10 µL) was added to each well and incubated for 3 h. Absorbance (optical density, OD) was measured at 570 nm with a microplate reader (Epoch, BioTek Instruments, Bad Friedrichshall, Germany); absorbance at 600 nm was used as a reference. Data were collected from two independent experiments with eight replicates per experiment and expressed as fold change versus vehicle-treated cells.

### 2.5. DNA Extraction

DNA from HepG2 and HCT116 cells was extracted with a Qiagen DNA MiniKit (Qiagen, Hilden, Germany) and a Qiacube (Qiagen), following the manufacturer’s instructions. Briefly, 2 µL of sample was pipetted into the plate spectrophotometer; elution buffer (2 µL) served as the blank. DNA quality and concentration were measured with a microplate spectrophotometer (Epoch, BioTek Instruments). DNA was measured using the Nucleic Acid Quantification tool (Gen5 software version 2.01, BioTek Instruments); only samples with 260/280 and 260/230 ratios above 1.8 were used in this study.

### 2.6. Quantification of Global DNA Methylation (5mC)

Global 5mC levels were measured colorimetrically using 100 ng DNA per sample with the MethylFlash Methylated DNA Quantification Kit (Epigentek, New York, NY, USA), following the manufacturer’s instructions. A microplate reader (Epoch, BioTek Instruments) was used to measure absorbance at 450 nm. Data are expressed as mean ± SD and expressed as percentage of 5 mC. We created a standard curve using linear regression (Microsoft Excel; Redmond, WA, USA) to determine the absolute quantity of methylated DNA. The amount and percentage of 5 mC were calculated as follows:5 mC (ng) = (Sample OD − Blank OD)/(Slope × 2)(1)
5 mC (%) = 5 mC (ng)/sample DNA (ng) × 100(2)

### 2.7. RNA Extraction

RNA from HepG2 and HCT116 cells was extracted with TRI Reagent (Sigma-Aldrich, St. Louis, MO, USA), following the manufacturer’s directions. DNA quality and concentration were measured with a microplate spectrophotometer (Epoch, BioTek instruments). Only RNA samples with 260/280 and 260/230 ratios above 1.8 were used in this study.

### 2.8. Quantitative Real-Time PCR (qPCR)

RNA was reverse-transcribed according to the specifications of the High Capacity cDNA Reverse Transcription Kit (Thermo Fisher). Purified RNAs (400 ng) were copied into cDNAs using gene-specific primers under the following thermocycling conditions: 25 °C (10 min), 37 °C (120 min), and 85 °C (5 min).

The StepOne Plus Real-Time PCR system (Thermo Fisher Scientific, Waltham, MA, USA) was used to quantify DNMT1 and DNMT3a expression and followed the manufacturer’s protocol. Each PCR reaction was performed in duplicate with the Taqman Gene Expression Master Mix (Thermo Fisher) and TaqMan probes (Thermo Fisher) for human DNMT3a (Assay ID Hs00945875_m1), DNMT1 (Assay ID Hs1027162_m1), SIRT1 (Hs0109006-m1), and SIRT2 (Hs01560289-m1). The comparative CT method was used to analyze the data [46] using the StepOne Plus Real-Time PCR software; data were expressed as fold-induction versus healthy samples. Data were normalized to human glyceraldehyde 3 phosphate dehydrogenase (GAPDH) (Assay ID Hs02786624_g1) mRNA levels. Data are shown as mean ± SD.

### 2.9. Sirtuin Activity Assay

#### 2.9.1. Nuclear Protein Extraction

Nuclear protein extracts from HepG2 and HCT116 cells were prepared using the EpiQuik Nuclear Extraction kit (Epigentek) following the manufacturer’s specifications. Protein concentrations of nuclear extracts were determined with the Pierce bicinchoninic acid (BCA) Protein Assay (Life Technology, Rockford, IL, USA).

#### 2.9.2. Quantification of SIRT Activity

SIRT activity was measured by a colorimetric SIRT Activity/Inhibition kit (Epigentek), as per the manufacturer’s instructions. Briefly, 100 ng nuclear protein extract was added to wells containing an acetylated histone-derived substrate and incubated for 90 min at 37 °C. The wells were rinsed with 1× wash buffer, and capture and detection antibodies were added. The amount of deacetylated product, proportional to SIRT enzyme activity, was then measured by recording the absorbance at 450 nm in a microplate spectrophotometer.

### 2.10. Statistical Analysis

Data were tested for normality and equality of variances using the Shapiro–Wilk and Levene’s tests, respectively. Statistical significance was determined with one-way ANOVA with post hoc Bonferroni correction for multiple comparisons with SPSS software (SPSS Inc., Chicago, IL, USA). Data are presented as mean ± S.E.M or S.D; * *p* < 0.05, ** *p* < 0.01 and *** *p* < 0.001 were considered statistically significant.

## 3. Results

### 3.1. Effect of AntiGan on Cell Viability of Human Cancer Cell Lines

To determine the effect of AntiGan on cancer cell viability, two different human cancer cell lines were selected. HepG2 is a hepatocarcinoma cell line derived from the liver of a 15-year-old Caucasian male with well-differentiated hepatocellular carcinoma [47]. HCT116, a human colon cancer cell line with epithelial morphology, is used in therapeutic research and drug screening [48]. HepG2 and HCT116 cells were treated with AntiGan (10, 50, 100, and to 500 µg/mL) for 24 h, 48 h, and 72 h (Figure 1). After 24 h of treatment, AntiGan had no effect on cell viability (Figure 1A). However, in the presence of 10 µg/mL AntiGan, cell viability decreased by approximately 30% after 48 h (Figure 1B). The effect of AntiGan on cell viability was not dose dependent, and higher concentrations of AntiGan produced no additive effect (Figure 1B). Similar results were obtained in both cell lines 72 h after AntiGan treatment (Figure 1C).

### 3.2. AntiGan Treatment Induces Apoptosis in Human Cancer Cell Lines

Bax is a member of the Bcl-2 family member with proapoptotic properties [1,49,50]. Given the role of Bax in apoptosis, we tested whether AntiGan induced any changes in Bax expression when added to the culture medium of HepG2 and HCT116 cells. In HepG2 and HCT116 cells, AntiGan (10 and 50 µg/mL) caused a high increase in Bax expression (Figure 2A,B), by 53–67% compared with vehicle-treated cells.

Tumor suppressor protein 53 (p53) is an important regulator of apoptosis [50]. Inactivation of p53 occurs in over 50% of all human cancers and causes the loss of a key DNA damage sensor, inducing an apoptotic effector cascade [1]. To analyze the effects of AntiGan on p53 expression, we treated HepG2 and HCT116 cell lines with 10 and 50 µg/mL AntiGan for 48 h and then immunostained these cells for p53. The p53 expression strongly increased, although not statistically significant, in AntiGan (50 µg/mL)-treated HepG2 cells compared with vehicle-treated cells (Figure 2A). The increase in p53 expression, caused by AntiGan treatment, was higher in HCT116 cells than in HepG2 cells (Figure 2B).

AntiGan displays cytotoxic and apoptotic activity in various human cell lines, including promyelocytic human leukemia (HL60), breast cancer (Hs 274.T), lung adenocarcinoma (H2126), melanoma (WM 115), and colorectal cancer (Caco-2, HT-29, SW-480), as well as in the dextran sulfate sodium (DSS) mouse model of colonic inflammation [34]. Our results confirm the proapoptotic property of AntiGan not only in a colorectal cancer cell line but also in a hepatocarcinoma cell line. To this, we analyzed the effect of AntiGan on cleaved caspase-3 expression but only in HepG2 cells. Caspases are crucial mediators of programmed cell death. Caspase-3 is a cell death protease associated with the formation of apoptotic bodies [51]. AntiGan (10 and 50 µg/mL) increased caspase-3 expression by 42–58%, respectively, with both concentrations of AntiGan (Figure 2A). Taken together, these data show that AntiGan treatment increases proapoptotic protein expression in HepG2 and HCT116 tumor cells.

### 3.3. AntiGan Regulates Cyclooxygenase-2 (COX-2) and Interleukin-17 (IL-17) Expression

COX-2 induces cancer stem cell-like activity and cancer cell-induced apoptotic resistance, proliferation, angiogenesis, inflammation, invasion, and metastasis [52]. AntiGan-treated HepG2 and HCT116 cells were analyzed for COX-2 immunoreactivity. AntiGan did not affect COX-2 expression in HepG2 cells (Figure 3A). AntiGan (10 µg/mL) produced a moderate (42%) decrease in HCT116 cells compared with untreated cells (Figure 3B). However, AntiGan (50 µg/mL) reduced COX-2 expression by 61% in HCT116 cells (Figure 3B).

Interleukin-17 (IL-17) is a proinflammatory cytokine involved in the formation, growth and metastasis of various cancers [53]. AntiGan (10 and 50 µg/mL) reduced IL-17 expression in HepG2 cells (Figure 3A). The effect of AntiGan was stronger in HCT116 cells; this decrease was 74% higher in cells treated with 50 µg/mL AntiGan than in 10 µg/mL AntiGan-treated cells, which showed a 39% increase in IL-17 expression (Figure 3B).

### 3.4. AntiGan Regulates DNA Methylation in HepG2 and HCT116 Cells

DNA hypomethylation is a ubiquitous feature in human tumors [54]. We tested the effect of AntiGan on global DNA methylation; 5mc levels increased in HepG2 and HCT116 cells exposed to AntiGan (Figure 4A,B). In both cell lines, this increase was higher in the presence of 10 µg/mL than 50 µg/mL; AntiGan (10 µg/mL) increased 5mC levels by 55% in HepG2 cells (Figure 4A) and by almost 50% in the HCT116 cell line (Figure 4B). AntiGan, therefore, reversed induction of hypomethylation in these tumor cells.

Given the increase in global DNA methylation caused by AntiGan treatment, we used qPCR to analyze DNMT expression in HepG2 and HCT116 cell lines. AntiGan (10 µg/mL) strongly reduced DNMT1 and DNMT3a expression in HepG2 and HCT116 cells compared with vehicle (Figure 4C,D). There were no differences in DNMT1 and DNMT3a expression between 10 and 50 µg/mL AntiGan. Therefore, AntiGan (10 µg/mL) was sufficient for regulating the DNA methylation machinery. We next evaluated DNMT protein expression after AntiGan treatment with immunofluorescence microscopy. DNMT3a protein levels, however, were higher in HepG2 and HCT116 cells after AntiGan treatment than in untreated cells (Figure 4G,H). This increase was observed at both concentrations of AntiGan (10 and 50 µg/mL).

However, detailed analysis showed differences in terms of DNMT3a localization. DNMT3a immunoreactivity was observed primarily within nuclei in vehicle-treated HepG2 and HCT116 cells. However, AntiGan decreased nuclear, but increased cytoplasmic, DNMT3a immunopositivity in both cell lines (Figure 5A,B). The presence of cytoplasmic DNMT3a staining after AntiGan treatment may explain the disparity in the effect of AntiGan between our mRNA and protein data.

### 3.5. AntiGan Regulates Sirtuin Expression and Activity in HepG2 and HCT116 Cells

Sirtuins regulate cancer cell survival, apoptosis, metastasis, and tumorigenesis [55]. AntiGan reduced SIRT activity in HepG2 and HCT116 cell lines (Figure 6). In HepG2 cells, AntiGan (50 µg/mL) caused a slight decrease in SIRT activity (*p* = 0.04) (Figure 6A); however, AntiGan, at concentrations of 10 and 50 µg/mL, significantly lowered SIRT activity in HCT116 cells only (*p* = 0.034, and *p* = and 0.041, respectively) (Figure 6B). While SIRT1 stimulates cell growth and proliferation in a variety of cancer subtypes [52], it may also act as a tumor suppressor [56]. AntiGan (10 and 50 µg/mL) reduced SIRT1 mRNA expression in HepG2 and HCT116 cells compared with untreated cells, and these reductions were similar at both concentrations of AntiGan (Figure 6C,D). SIRT1 mRNA levels decreased from 1 to 0.0067 (AntiGan, 10 µg/mL) and 0.01 (AntiGan, 50 µg/mL) in HepG2 cells (Figure 6C). In HCT116 cells, SIRT1 mRNA levels expression decreased from 1 to 0.086 (AntiGan, 10 µg/mL) and 0.068 (AntiGan, 50 µg/mL), respectively. SIRT2 acts as a tumor suppressor and an oncogene [55]; it interacts with β-catenin and lysine-specific demethylase 4A (KDM4A) to inhibit cell growth, and is involved in survival and cell proliferation in pancreatic cancer, hepatocarcinoma, and neuroblastoma [55,56]. AntiGan (10 and 50 µg/mL) reduced SIRT2 mRNA levels over 10-fold in HepG2 cells (from 1 to 0.053 and 0.057, respectively) (Figure 6E). AntiGan (10 and 50 µg/mL) lowered SIRT2 levels by approximately 70% (from 1 to 0.35 and 0.38, respectively) in HCT116 cells (Figure 6F). These reductions were again similar at both concentrations of AntiGan. To determine the effect of AntiGan on SIRT1 protein expression, we immunolabeled HCT116 cells with an antibody against SIRT1; immunopositive SIRT1 cells were slightly more numerous in the presence of 10 and 50 µg/mL AntiGan than in untreated cells (Figure 6G).

Immunolocalization of SIRT1 was altered in tumor cells treated with AntiGan. In HCT116 cells exposed to AntiGan (10 µg/mL), SIRT1 immunoreactivity was detected in the nuclei and cytoplasm; in vehicle-treated cells, SIRT1-positive staining was confined mainly within nuclei. Cytoplasmic SIRT1 localization was higher in HCT116 cells treated with 50 µg/mL AntiGan (Figure 7).

## 4. Discussion

Current cancer treatments, such as chemotherapy, radiotherapy, and surgery, have side effects that compromise patient health and well-being. Using nutraceuticals to treat prevalent diseases has increased in recent years [25,28,57,58]. In addition to their nutritional value, biological compounds extracted from natural sources may be beneficial to human health [27]. Nutraceuticals have the potential to slow cancer cell growth, inhibit proliferation, and induce cancer cell apoptosis, angiogenesis, and metastasis [59,60].

For over three decades, one of the major goals of clinical oncology has been the development of apoptosis-based drugs. This programmed cell death process is mediated by several signaling pathways and factors [1,49,50]. AntiGan inhibits growth in several colorectal cancer cell lines and regulates apoptosis-related p53, p21, Bax, and Bcl-2 gene expression [34]. In an in vivo colitis-associated dysplasia and tumor mouse model, Bcl-2 expression was downregulated after treatment with AntiGan [34]. AntiGan exhibits antitumoral properties [31,33,34], but the molecular or epigenetic mechanisms underlying this effect have not been identified. In the present study, AntiGan reduced the viability/survival of human liver and colon cancer cells in vitro; AntiGan increased apoptosis in both cancer cell lines by regulating p53 and Bax and cleaved caspase-3 protein expression, suggesting that AntiGan is an effective chemotherapeutic compound in treating various types of tumors (Figure 8).

COX-2 promotes carcinogenesis in several cancer types and is critical in cancer cell resistance to chemo- and radiotherapy [52]. Natural compounds regulate COX-2 expression and are a promising tool for cancer chemoprevention and therapy [61]. AntiGan reduces COX-2 expression in crypt cells of the middle and distal colon in DSS-treated mice [34]. In the current study, AntiGan reduced COX-2 expression in HepG2 and HCT116 cells in culture. AntiGan, in addition to being protective against colorectal cancer, may therefore also be therapeutically beneficial against other types of cancers. Furthermore, AntiGan decreased IL-17 production in HepG2 and HCT116 cells. A progressive increase in IL-17 mRNA levels correlates with severe dysplasia [62]. Human hepatocarcinoma contains a high proportion of IL-17-positive cells, and elevated IL-17 levels are associated with microvessel density and poor survival [63]. Our group previously showed that AntiGan treatment reduces the density of different tumor markers in healthy subjects (*n* = 50) and in patients with colon, breast, prostate, and lung cancer (*n* = 156); this effect of AntiGan was more evident in cancer patients [35]. In that same study, AntiGan reduced alpha-fetoprotein (AFP) and carcinoembryonic antigen (CEA) levels; AFP is a marker for hepatocellular carcinoma and CEA is a marker for gastrointestinal cancer [35].

Cancer cells exhibit aberrant hypermethylation and global genomic hypomethylation, which contributes to their highly proliferative and apoptosis-resistant phenotype. In the current study, AntiGan increased global DNA methylation, suggesting that it acts as an antitumoral epidrug. Epigenetic therapy is therapeutically beneficial against various types of cancers, and epidrugs sensitize cancer cells when used in combination with standard chemotherapy [6,64,65,66]. DNMT1 inhibitors such as azacitidine and decitabine cause progressive hypomethylation, increasing the expression of tumor suppressor genes [26]. Moreover, azacytidine increases methylation of CpGs implicated in the response of HCT116 cells to decitabine [67]. However, toxicity issues and off-target effects of these drugs limit their widespread use.

Apoptosis is regulated by genetic and epigenetic factors. Resistance to apoptosis is aided by DNA demethylation, which modulates the apoptotic response of tumor cells [58]. Hypermethylation blocks the initiation and progression of intrinsic and extrinsic apoptosis pathways [60]. Mutations in DNMT1 and DNMT3a overexpression are features of both cancer patients and a mouse model of colon tumors [64]; increased DNMT1 and DNMT3a expression contribute to hepatocellular carcinogenesis [64]. Nuclear expression of DNMT1 is elevated in hepatocarcinoma and dysplastic nodules [65]. In the present study, AntiGan reduced DNMT1 and DNMT3a mRNA expression, suggesting that this nutraceutical has antitumoral epigenetic properties. DNMT3a is overexpressed in gastric cancer tissues, where it localizes to the cell cytoplasm [66]. DNMT3a cytoplasmic immunoreactivity is found in non-neoplastic and dysplastic nodules and hepatocarcinoma [65]. Nuclear DNMT3a is undetectable in non-neoplastic liver and low-grade dysplastic nodules but occurs in high-grade dysplastic nodules and hepatocarcinoma [67]. AntiGan treatment increases DNMT3a immunoreactivity; this is mostly localized to the cytoplasm, suggesting that this treatment may slow cancer progression.

High SIRT1 expression is found in several cancer cell lines and is associated with poor prognosis and survival [68]. SIRT1 promotes cell survival by deacetylating p53 and inhibiting its function [69]; its expression in liver cancer correlates with poor prognosis and low survival rates [70]. Chemoresistance in SIRT1-overexpressing tumors causes cancer cell hyperproliferation and survival [70], and SIRT1 contributes to colon cancer initiation, invasion, and metastasis [71]. SIRT1 may facilitate cancer progression by epigenetically modulating the polycomb repressor complex 4 (PRC4) comprising DNMT1, DNMT3b, polycomb group (PcG) proteins, embryonic ectoderm development isoform 2 (EED2), and enhancer of zeste homolog 2 (EZH2) [72]. Our data show that AntiGan reduces SIRT1 mRNA levels, suggesting that it acts as an epinutraceutical against liver and colon tumor. However, SIRT1 functions as a tumor suppressor but is also involved in tumorigenesis [68].

SIRT1, localized to the cytoplasm, suppresses the mesenchymal program, activates the epithelial program, and inhibits tumoral cell migration and invasion [73,74]. Our immunofluorescence data show that SIRT1 is localized both in nucleus and cytoplasm in HCT116 cells. AntiGan increased SIRT1 immunoreactivity in the cytoplasm in HCT116 cells in a concentration-dependent manner. These data suggest that AntiGan may reduce the metastatic potential of HCT116 cells. However, additional research is required to determine whether AntiGan prevents metastasis formation. Indeed, this change in the SIRT1 cellular localization pattern may explain the opposite results observed between SIRT1 activity and mRNA levels and protein expression, where increased protein expression may be caused by the increment in SIRT1 levels in the cytoplasm but not in the nucleus.

SIRT2 regulates mitosis, cell motility, differentiation, oxidative metabolism, and cell death [68], and it has tumor suppressor and oncogenic activities during tumorigenesis. In hepatocarcinoma, SIRT2 is upregulated and promotes vascular invasion, cell proliferation, and tumor growth [71]. SIRT2 knockdown or its pharmacological inhibition inhibits cancer cell proliferation and growth [71,75]. Novel SIRT2 inhibitors potently inhibit tumor growth in a HCT116 xenograft murine model, supporting a role for SIRT2 as a therapeutic target for colorectal cancer [76]. Taken together, those findings support our present data; that is, AntiGan reduced SIRT mRNA levels, showing that AntiGan exerts an antitumoral and epigenetic effect in HCT116 and HepG2 cells. SIRT2 knockdown experiments in mice, however, reveal a robust tumor suppressor role for SIRT2 [70]. This is contradictory and may be attributed to the use of different models at different stages of tumor development. More research is needed to determine the precise role of sirtuins in carcinogenesis.

Nicotinamide, a water-soluble form of vitamin B3, inhibits SIRT1, SIRT2, SIRT3, SIRT5, and SIRT6 expression [17]. Nicotinamide, furthermore, prevents human prostate cancer cell growth and survival [70]. Benzamide, a nicotinamide mimic, is a potent and selective SIRT2 inhibitor that increases α-tubulin acetylation in HCT116 cells [77]. MHY2256 is a novel SIRT inhibitor that induces cell cycle arrest, apoptosis, and autophagic cell death in HCY116 cells [78]. Our data show that AntiGan reduced SIRT activity and SIRT1 and SIRT2 mRNA levels. AntiGan may therefore act as a SIRT inhibitor and as an epinutraceutical to treat cancer; this lipofishin shows epigenetic and antitumoral effects, and as a naturally derived compound, would exhibit fewer side effects than those observed following traditional chemotherapy.

The cooperative actions of HDACi- and DNMTi-compounds suggest that combinatorial drug administration may be an attractive clinical strategy against cancer. Several clinical trials are being conducted using a combination of both types of inhibitors [79]. The current study demonstrates that AntiGan inhibits DNMTs and SIRTs, emphasizing its potential as a drug for epigenetic therapy. AntiGan is a promising epinutraceutical that targets cancer-related epigenetic mechanisms.

## 5. Conclusions

In this study, we evaluated AntiGan as a potential therapeutic agent against cancer. AntiGan increased apoptosis and reduced cell viability and COX-2 and IL-17 expression in HepG2 and HCT116 cell lines. Furthermore, AntiGan regulated DNA methylation and SIRT activity and expression, indicating that AntiGan is an epidrug against tumor cells. Although several epidrugs have been developed and tested in clinical trials for treating tumors, they were associated with systemic toxicity and side effects. Nutraceuticals, however, may be an efficient and reliable treatment for avoiding such undesirable effects in patients with cancer. Our data show that AntiGan is a powerful antitumor and epigenetic compound against liver and colon cancer cells in vitro; these results may help expand current and future therapeutic options when assessing treatment of patients with cancer.

## Figures and Tables

**Figure 1 life-12-00097-f001:**
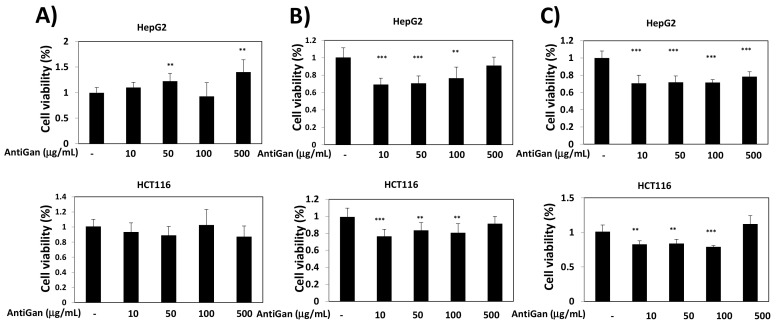
AntiGan reduces cell viability in cancer cell lines. HepG2 and HCT116 cells were treated with AntiGan (10–500 µg/mL); results after 24 h (**A**), 48 h (**B**), and 72 h (**C**). Cell viability was determined with the PrestoBlue Cell Viability assay. Eight replicates of each condition were performed, and results are represented as fold change compared with vehicle-treated cells. Data are expressed as mean ± S.E.M. ** *p* < 0.01; *** *p* < 0.001.

**Figure 2 life-12-00097-f002:**
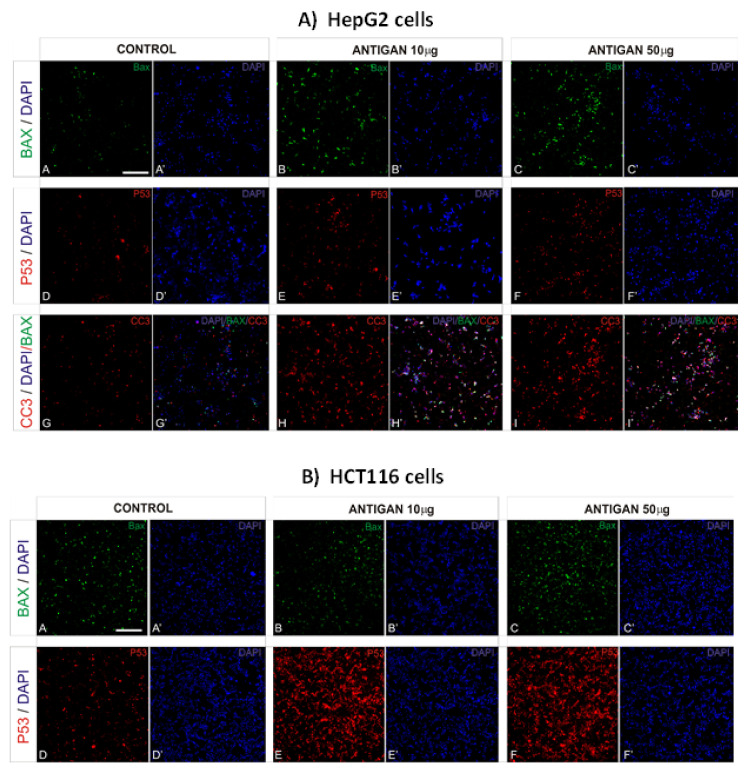
AntiGan induces apoptosis in cancer cell lines. HepG2 (**A**) and HCT116 (**B**) cells were treated with AntiGan (10 and 50 µg/mL) for 48 h and then immunostained with antibodies against Bax (green), p53 (red), and CC3 (red). A, A’, D, D’, G, G’, Control cells; B, B’, E, E’, H, H’, ANTIGAN 10 ug/mL; C, C’, F, F’, I, I’, ANTIGAN 50 ug/mL. Nuclei were labeled with 4′,6-diamidine-2′-phenylindole dihydrochloride (DAPI, blue). Apostrophe indicates DAPI staining CC3, cleaved caspase-3. Scale bar, 100 µm for all images.

**Figure 3 life-12-00097-f003:**
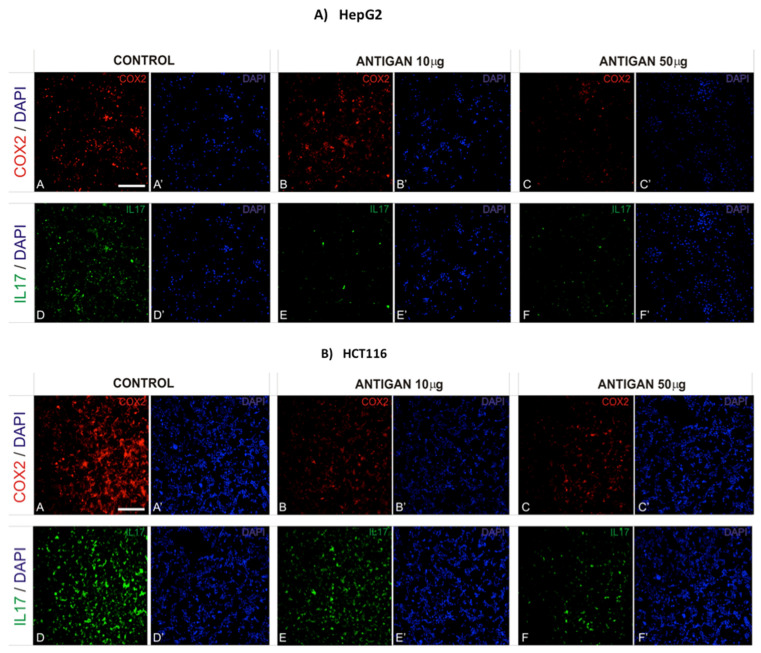
AntiGan regulates COX-2 and IL-17 expression in cancer cells in vitro. HepG2 (**A**) and HCT116 (**B**) cells were treated with AntiGan (10 and 50 µg/mL) for 48 h. The cells were immunolabeled with antibodies against COX-2 (red) and IL-17 (green). A, A’, D, D’, Control cells; B, B’, E, E’, ANTIGAN 10 ug/mL; C, C’, F, F’, ANTIGAN 50 ug/ml. Cells were counterstained with DAPI (blue) to label nuclei. Apostrophe indicates DAPI staining COX-2, cyclooxygenase-2; IL-17, interleukin 17. Scale bar, 100 µm for all images.

**Figure 4 life-12-00097-f004:**
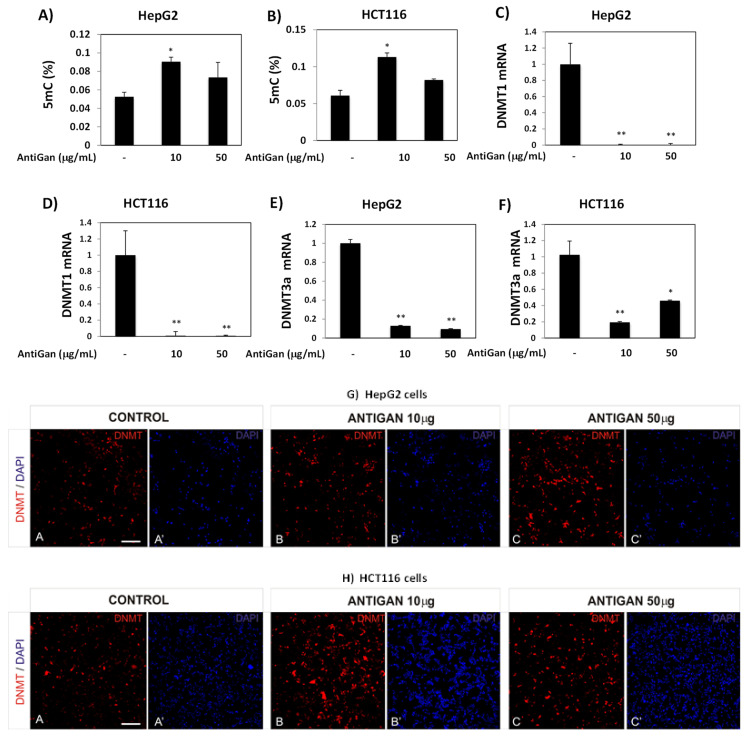
AntiGan regulates DNA methylation in liver and colorectal cancer cell lines. HepG2 and HCT116 cells were treated with AntiGan (10 and 50 µg/mL) for 48 h. Global DNA methylation (5mC) levels were measured in HepG2 (**A**) and HCT116 (**B**) cells and expressed as percentages. DNMT1 mRNA levels were measured by qPCR in HepG2 (**C**) and HCT116 (**D**) cells. DNMT3a mRNA levels were also measured by qPCR in HepG2 (**E**) and HCT116 (**F**) cells. Results are expressed as fold-change compared with untreated cells. HepG2 (**G**) and HCT116 (**H**) cells were immunostained with an antibody against DNMT3a (red). A, A’ Control cells; B, B’ ANTIGAN 10 ug/mL; C, C’ ANTIGAN 50 ug/mL. DAPI (in blue) was used to mark nuclei. Apostrophe indicates DAPI staining. Data are expressed as mean ± S.E.M. Statistical significance between groups was calculated with one-way ANOVA with post hoc Bonferroni correction for multiple comparisons and it is shown as * *p* < 0.05 and ** *p* < 0.01. 5mC, 5-methylcytosine; qPCR, quantitative real-time PCR. Scale bar, 100 µm for all images.

**Figure 5 life-12-00097-f005:**
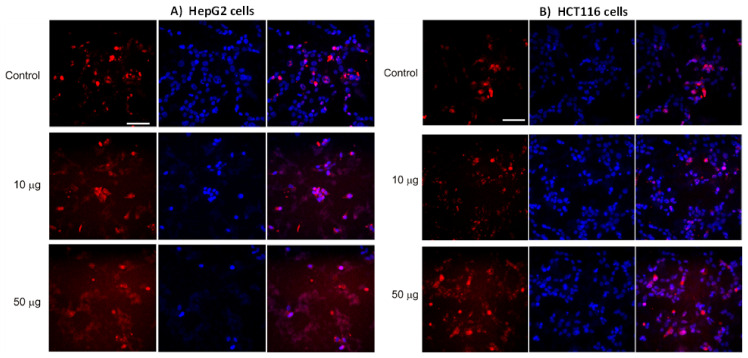
AntiGan regulates the cellular localization of DNMT3a in cancer cells in vitro. HepG2 (**A**) and HCT116 (**B**) cells were treated with AntiGan (10 and 50 µg/mL) for 48 h. The cells were immunostained with an antibody against DNMT3a (red). Cells were counterstained with DAPI (blue) to label nuclei. Merged images showing DNMT3a immunoreactivity/DAPI staining are in the right column of panels for the HepG2 and HCT116 cell lines. Scale bar, 100 µm for all images.

**Figure 6 life-12-00097-f006:**
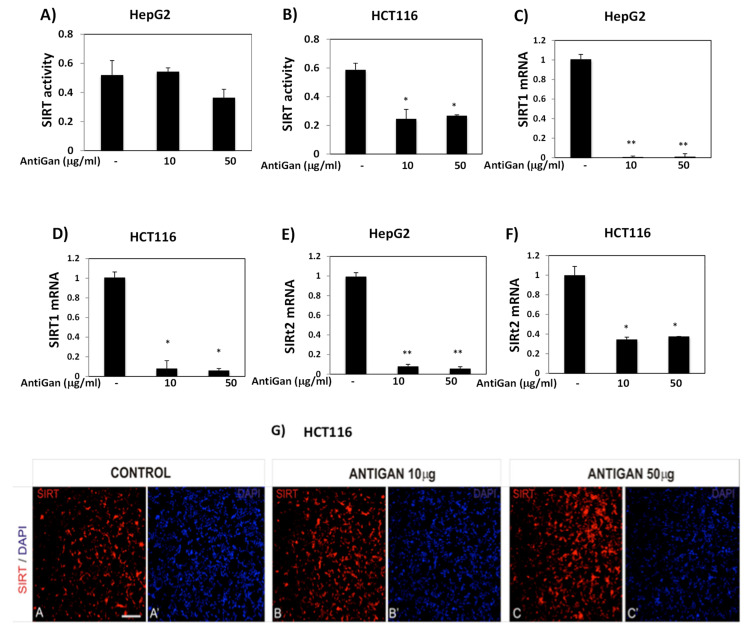
AntiGan regulates SIRT activity and expression. HepG2 and HCT116 cells were treated with AntiGan (10 and 50 µg/mL) for 48 h. Sirtuin activity was measured in HepG2 (**A**) and HCT116 (**B**) cells. SIRT1 mRNA levels were measured by qPCR in HepG2 (**C**) and HCT116 (**D**) cells. SIRT2 mRNA levels were also measured by q in HepG2 (**E**) and HCT116 (**F**) cells. Results are expressed as fold induction versus untreated cells. (**G**) HCT116 cells were immunolabeled with an antibody against SIRT1 (red). A, A’ Control cells; B, B’ ANTIGAN 10 ug/mL; C, C’ ANTIGAN 50 ug/mL Cells were counterstained with the nuclear DNA marker DAPI (blue). Apostrophe indicates DAPI staining. Statistical significance between groups was calculated with one-way ANOVA with post hoc Bonferroni correction for multiple comparisons and it is shown as * *p* < 0.05 and ** *p* < 0.01. qPCR, quantitative real-time PCR; SIRT, sirtuin.

**Figure 7 life-12-00097-f007:**
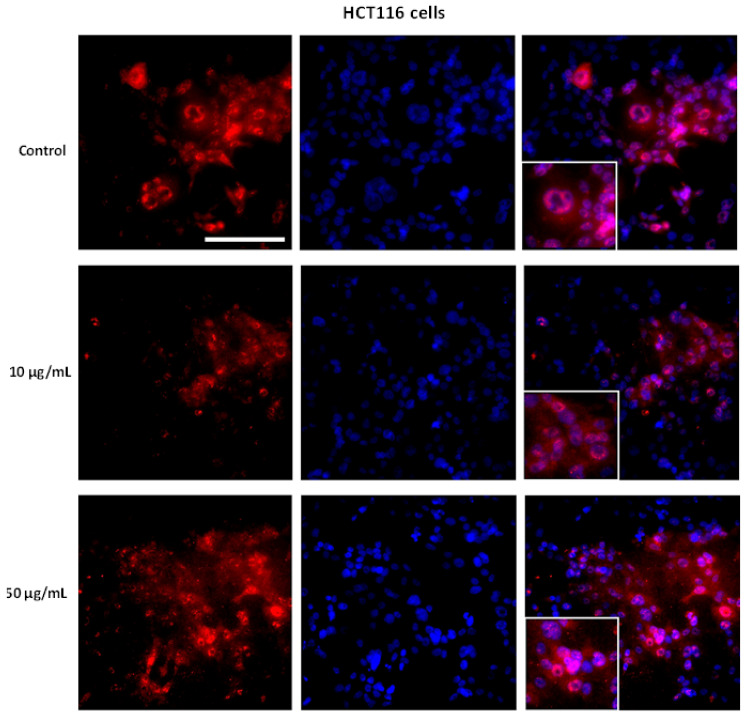
AntiGan regulates SIRT1 cellular localization. HCT116 cells were treated with treated with AntiGan (10 and 50 µg/mL) for 48 h. The cells were immunolabeled with an antibody against SIRT1. Nuclei were counterstained with DAPI (blue). Merged images showing SIRT1 immunoreactivity/DAPI staining are in the right column of panels for HCT 116 cells. Scale bar, 150 µm for all images.

**Figure 8 life-12-00097-f008:**
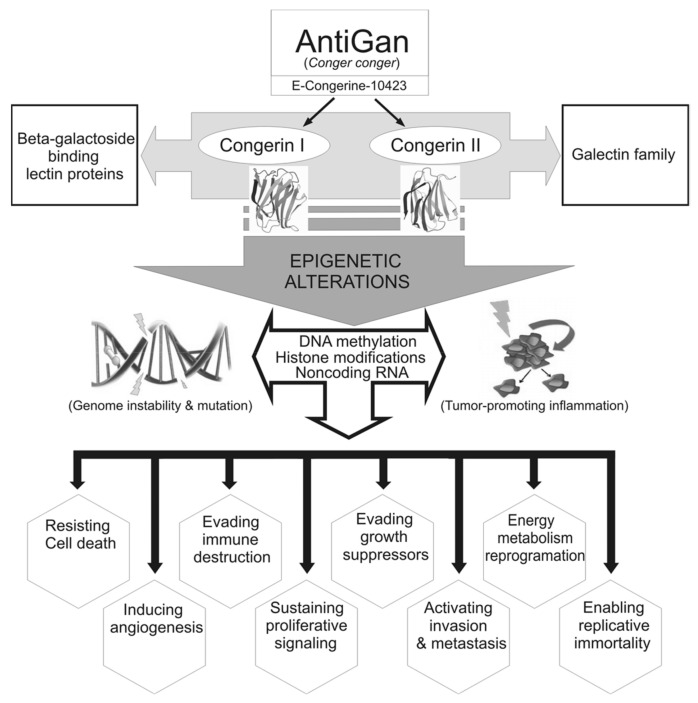
The central role of epigenetic mechanisms induced by AntiGan against cancer hallmarks. Epigenetic modifications such as DNA methylation, histone modifications, and noncoding RNAs may be reshaped by galectin proteins (congerin I and II) as key factors to ameliorate pathological hallmarks in cancer. These features are influenced by processes such as genomic instability and mutation and tumor-promoting inflammation, which in turn lead to alterations in epigenetic mechanisms. AntiGan increased apoptosis and reduced cell viability and COX-2 and IL-17 expression in hepatocellular carcinoma (HepG2) and colorectal carcinoma (HCT116) cell lines. AntiGan also regulated DNA methylation and SIRT activity and expression, indicating that AntiGan is an epidrug against tumor cells. COX-2, cyclooxygenase-2; IL-17, interleukin-17; SIRT, sirtuin.

**Table 1 life-12-00097-t001:** Composition of E-Congerine-10423 extract—the structural base of AntiGan.

** COMPOSITION (%) **			** MINERALS per 100 g **	
PROTEINS	75–85		PHOSPHORUS	100 mg
LIPIDS	0.5–1.5		CALCIUM	164 mg
CARBOHYDRATES	0.4		MAGNESIUM	134 mg
** AMINO ACIDS: % IN PROTEINS **			IRON	14 mg
GLUTAMIC ACID	13.71		ZINC	3 mg
ASPARTIC ACID	8.26		** VITAMINS per 100 g **	
LYSINE	7.80		VITAMIN B_3_	760 mg
LEUCINE	6.49		VITAMIN B_1_	0.2 mg
ARGININE	5.37		VITAMIN B_2_	0.08 mg
ALANINE	4.76		VITAMIN D	0.25 mcg
VALINE	3.82		** LIPIDS **	
THREONINE	3.76		**SATURATED FATTY ACIDS: % IN LIPIDS**	
ISOLEUCINE	3.70		PALMITIC	21.8
SERINE	3.35		STEARIC	8.60
PHENYLALANINE	3.21		MYRISTIC	4.60
GLYCINE	3.10		**POLYUNSATURATED FATTY ACIDS: % IN LIPIDS**	
PROLINE	2.90		OLEIC	23.20
TYROSINE	2.71		PALMITOLEIC	5.50
METHIONINE	2.29		GADOLEIC	4.90
HISTIDINE	2.09		**MONOSATURATED FATTY ACIDS: % IN LIPIDS**	
CYSTEINE	0.94		LINOLEIC	12.00
TRYPTOPHAN	0.84		DHA	6.30
			LINOLENIC	2.00
			EPA	1.80

**Table 2 life-12-00097-t002:** List of antibodies for immunocytochemistry.

Antibody	Species	Clonality	Supplier	Product Number	Ref.
Bax	Mouse	Monoclonal	Thermo Fisher	MA5-14003	[39]
IL-17	Mouse	Monoclonal	Thermo Fisher	12-7177-81	[40]
p53	Rabbit	Monoclonal	Thermo Fisher	MA5-14516	[41]
Cleaved caspase-3	Rabbit	Polyclonal	Cell Signaling	9664	[42]
Cox-2	Rabbit	Monoclonal	Vector Labs.	VP-RM02	[43]
Dnmt3a	Rabbit	Polyclonal	Abcam	ab2850	[44]
Sirt1	Rabbit	Monoclonal	Abcam	ab32441	[45]

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
