# Peer review of "AntiGan: An Epinutraceutical Bioproduct with Antitumor Properties in Cultured Cell Lines"

_life, 2022, doi:10.3390/life12010097_

Round 1

Reviewer 1 Report

In this paper, Martínez-Iglesias et al. investigated the epigenetic effects of the nutraceutical product AntiGan (a lipofishin extract from the sea eel Conger conger) cancer cell lines. Specifically, the authors assessed the cytotoxicity of AntiGan in HepG2 and HCT116 cells and the effects on the expression and localization of several proteins including Bax, p53, COX-2, IL-17, cleaved caspase-3 as well selected epigenetic enzymes (DNMT1/3A, SIRT1/2) using qRT-PCR and immunocytochemistry (ICC). Overall, the paper provides a description of the molecular and epigenetic effects of AntiGan in selected cell lines and the authors suggest it can potentially serve as an epinutraceutical with antitumor properties.

Major comments:

  1. The rationale for the investigation of epigenetic effects of AntiGan is not very clear. The authors need to clarify why they hypothesized AntiGan may specifically serve as epinutraceutical among other biological effects.
  2. The assumption that nutraceuticals, despite their poorly defined composition, are generally safer that synthetic and semi-synthetic drugs may not be very true, especially if they exert similar pharmacological effects. Perhaps the authors need to rephrase sections that build on this assumption.
  3. The authors investigated discrete molecular changes that are not clearly connected in the paper. The authors need to provide a coherent model of how AntiGan, with its multiple bioactive ingredients, elicits these broad cellular effects. A working model schematic may help with this clarification.
  4. The primer sequences used for qRT-PCR need to be provided
  5. The authors need to provide ICC images with higher magnification and resolution, particularly with those illustrating localization to be able to clearly observe changes in the localization.
  6. AntiGan-induced reductions in DNMT1, DNMT3a, SIRT1, SIRT2 expression as assessed by qRT-PCR are too profound to be linked with the slight reductions in viability. Even if these are not essential for viability, these epigenetic effects are significant and thus the authors will need to support these data with immunoblotting.

Minor comments:

Scale bars for ICC images need to be provided

Author Response

REVIEWER 1

Major Comment 1:

The rationale for the investigation of epigenetic effects of AntiGan is not very clear. The authors need to clarify why they hypothesized AntiGan may specifically serve as epinutraceutical among other biological effects.

Response:

Thank you to the reviewer. We have addressed the reviewer’s question in the Introduction on page 3 (lines 118-130).

Several drugs that target different components of the epigenetic apparatus for the treatment of cancer have, in recent years, entered clinical practice, and others are being evaluated in clinical trials. For example, the DNA methyltransferase 1 (DNMT1) inhibitor 5-azacytidine is FDA-approved for the treatment of  myelodysplastic syndromes; the histone deacetylase (HDAC) inhibitor panobinostat, also FDA-approved, is used to treat multiple myeloma; another histone deacetylase (HDAC) inhibitor entinostat is in clinical trials for the treatment of kidney-, colon-, and breast cancer, and Hodgkin lymphoma; and, the FDA-approved histone methyltransferase EZH2 inhibitor tazemetostat is used to treat follicular lymphoma.

Previously published findings by our group with AntiGan showed four major findings:

  • AntiGan produces cytotoxicity in the following tumor cell lines of human origin: HL-60 (acute promyelocytic leukemia), HS 274.T (adenocarcinoma of the breast), HS313.T (lymphoma), H2126 (non-small cell lung adenocarcinoma), WM 115 (melanoma), and Caco-2 (colorectal adenocarcinoma), HT-29 (colorectal adenocarcinoma) and SW-480 (colorectal adenocarcinoma) (Lombardi et al. 2018)
  • AntiGan, in an in vivo rat model of cold stress stimulates the immune system by increasing the expression of CD4 and CD28 antigens in T lymphocytes, and granulocyte- and monocyte phagocytic activities (Lombardi et al. 2006).
  • AntiGan, in an in vivo mouse model of ulcerative colitis, protects against the development of premalignant lesions (Lombardi et al. 2018).
  • Clinically, daily administration of AntiGan (750 mg) for one month to patients (n = 156) with colon, breast, prostate or lung cancer, reduces serum levels of tumor markers (alpha-fetoprotein, alpha-amylase, prostate-specific antigen, lipase, carcinoembryonic antigen, prostate-specific antigen) in approximately 50% of these patients (Lombardi et al. 2019).

These studies, however, did not test the potential for AntiGan as a nutraceutical product with epigenetic properties (epinutraceutical). Given that AntiGan exhibits anti-tumor properties, the aim of the present study was to determine, for the first time, whether AntiGan act as an epinutraceutical by regulating the epigenetic machinery that drives liver and colon cancer development and progression.

References:

   Lombardi, V.R.M.; Pereira, J.; Etcheverría, I.; Fernández-Novoa, L.; Seoane, S.; Cacabelos, R. Improvement of immune function by means of Conger conger extract in an in vivo rat model of cold stress. Food Agric. Immunol. 2006, 17, 115–127.

   Lombardi, V.R.M.; Carrera, I.; Cacabelos, R. In vitro and in vivo cytotoxic effect of AntiGan against tumor cells. Exp. Ther. Med. 2018, 15, 2547–2556.

   Lombardi, V.R.M.; Carrera, I.; Corzo, L.; Cacabelos, R. Role of bioactive lipofishins in prevention of inflammation and colon cancer. Semin. Cancer Biol. 2019, 56, 175–184

Major Comment 2:

The assumption that nutraceuticals, despite their poorly defined composition, are generally safer that synthetic and semi-synthetic drugs may not be very true, especially if they exert similar pharmacological effects. Perhaps the authors need to rephrase sections that build on this assumption.

Response:

We have removed those sentences concerning the safety of nutraceuticals, and have furthermore made the following change:

In the Abstract: the original sentence “Given the general absence of toxicity in nutraceuticals, targeting of the epigenetic apparatus with naturally derived bioactive compounds may be invaluable for cancer prevention and treatment” has been rewritten as “Targeting the epigenetic apparatus with bioproducts may aid cancer prevention and treatment.” (lines 15 and 16)

Major Comment 3:

The authors investigated discrete molecular changes that are not clearly connected in the paper. The authors need to provide a coherent model of how AntiGan, with its multiple bioactive ingredients, elicits these broad cellular effects. A working model schematic may help with this clarification.

Response:

Thank you to the reviewer. A working model (Figure 8; page 15) depicting the cellular effects of AntiGan has been added to the Discussion section in the revised manuscript. (lines 417–423).

Major Comment 4:

The primer sequences used for qRT-PCR need to be provided

Response:

TaqMan probes for qPCR gene expression assays were purchased from Thermo Fisher Scientific. Unfortunately, it is not possible to provide the primer sequences as this information is proprietary to Thermo Fisher and we do not have them. We have, however, provided Assays IDs for each TaqMan probe used (section 2.8. Quantitative Real-Time PCR (qPCR); lines 215–216).

Major Comment 5:

The authors need to provide ICC images with higher magnification and resolution, particularly with those illustrating localization to be able to clearly observe changes in the localization.

Response:

Thank you to the reviewer. We have provided new photomicrographs for each Figure. For each updated figure, the brightness has been intensified and the resolution increased from 300 points per pixel to 600 points per pixel. We have added a scale bar to the first image in an image-set; this scale bar applies to all images for that Figure.

Major Comment 6:

AntiGan-induced reductions in DNMT1, DNMT3a, SIRT1, SIRT2 expression as assessed by qRT-PCR are too profound to be linked with the slight reductions in viability. Even if these are not essential for viability, these epigenetic effects are significant and thus the authors will need to support these data with immunoblotting.

Response:

We do not, unfortunately, have HepG2 and HCT116 cells growing in culture at this time, or have samples processed for Western blotting experiments. Given the limited (10-day) timeframe for manuscript resubmission, it is not possible for us to presently conduct Western blotting, since we would first need to culture and then treat the cells. We will, furthermore, need to procure corresponding antibodies for DNMT1 and SIRT2. We appreciate this comment, however, and agree that these are important experiments to perform. Our future objective is to investigate their protein expression by Western blotting and ELISAs.

DNA methylation affects not only genes involved in cell viability but also those linked to angiogenesis, invasion, immune responses, and cellular signaling (Moyota M et al, 2011). Therefore, DNMT downregulation caused by AntiGan therapy may affect these cancer-related processes. In a similar way, reductions in SIRT1 and SIRT2 mRNA may have an impact on tumor-related processes such as inflammation (Kleszck R et al, 2021) and epithelial-mesenchymal transition (EMT) (Palmirotta R et al, 2016).

Minor Comments:

Scale bars for ICC images need to be provided

Response:

We have added scale bars (100 µm) to ICC images. The corresponding magnifications have now been stated in each Figure legend in the revised manuscript.

Reviewer 2 Report

This manuscript by Martínez-Iglesias et al. presents AntiGan: an epinutraceutical bioproduct with anti-tumor . I have some considerations for the authors:

  • In the abstract “Given the general absence of toxicity in nutraceuticals”. The authors claim that every nutraceutical has no toxicity. This can be stated if all toxicity tests are carried out. I recommend rephrasing this sentence.
  • In line 21 “To determine whether Anti- 20 Gan, a lipofishin extract from the sea eel Conger conger”, What does the word “lipofishin” mean?
  • The authors do not cite in the abstract, the inhibitory concentrations for each cell type. The abstract is very poor in results. It needs to show the results found in the work.
  • In line 106 “Cells were then exposed to 10µg/ml and 50 µg/ml AntiGan for 48 h”.

Why was it only evaluated in two concentrations? Why were these concentrations selected? Why wasn't it evaluated in 24 hours?

  • In line 129, “Cell viability was determined using PrestoBlue Cell Viability assay (Thermo Fisher 127Scientific, Massachusetts, USA). Cells (1 × 104) were incubated in 96-well plates with Anti-Gan (10, 50, 100 and 500 µg/mL)”. The evaluated concentrations are too high
  • The authors do not show the chemical composition of the evaluated extract. This information needs to be shown in the manuscript.

Author Response

REVIEWER 2

Comment 1:

In the abstract “Given the general absence of toxicity in nutraceuticals”. The authors claim that every nutraceutical has no toxicity. This can be stated if all toxicity tests are carried out. I recommend rephrasing this sentence.

Response:

Thank you to the reviewer. In the Abstract, the original sentence “Given the general absence of toxicity in nutraceuticals, targeting of the epigenetic apparatus with naturally derived bioactive compounds may be invaluable for cancer prevention and treatment” has been rewritten as “Targeting the epigenetic apparatus with bioproducts may aid cancer prevention and treatment.” (lines 15 and 16).

In the main text, the original sentence “Given the general absence of toxicity in nutraceuticals, epigenetic therapy, specific targeting of the epigenetic apparatus with naturally derived bioactive compounds, may aid cancer prevention and treatment”, has been removed from the revised manuscript.

Comment 2:

In line 21 “To determine whether Anti- 20 Gan, a lipofishin extract from the sea eel Conger conger”, What does the word “lipofishin” mean?

Response:

LipoFishins (LFs), are a new class of lipoproteins obtained from the muscle of different fish species, and are members of a novel category of nutraceutical compounds from the ProteoLipin family, a series of complex lipoproteins derived from marine resources.

References:

   Cacabelos, R.; Lombardi, V.; Fernández-Novoa, L.; Carrera, I.; Cacabelos, P.; Corzo, L.; Carril, J.C.; Teijido, O. Basic and clinical studies with marine lipofishins and vegetal favalins in neurodegeneration and age-related disorders. In Studies in Natural Products Chemistry; Atta-ur-Rahman, Ed.; Elsevier: Amsterdam, The Netherlands, 2018; Volume 59, pp. 195–225

   Cacabelos, R. Novel Biotechnological products from natural sources: Nutri/pharmacogenomic component. J. Nutr. Food Sci. 2016, 6, 6.

Comment 3:

The authors do not cite in the abstract, the inhibitory concentrations for each cell type. The abstract is very poor in results. It needs to show the results found in the work.

Response:

We have rewritten the Abstract and incorporated the changes required by the reviewer in the revised manuscript. The Abstract now includes information concerning the inhibitory drug concentrations of AntiGan for each cell type, relative to the markers analyzed.

Comment 4:

In line 106 “Cells were then exposed to 10µg/ml and 50 µg/ml AntiGan for 48 h”. Why was it only evaluated in two concentrations? Why were these concentrations selected? Why wasn't it evaluated in 24 hours?

Response:

Thank you to the reviewer for this comment. We used a cell viability assay to select AntiGan concentrations and treatment times. In our cell viability experiments, we selected lower doses of AntiGan that demonstrated activity. We did not conduct these studies at 24 hours because we did not observe any effect of AntiGan on cell viability at that timepoint.

Comment 5:

In line 129, “Cell viability was determined using PrestoBlue Cell Viability assay (Thermo Fisher 127Scientific, Massachusetts, USA). Cells (1 × 104) were incubated in 96-well plates with Anti-Gan (10, 50, 100 and 500 µg/mL)”. The evaluated concentrations are too high.

Response:

Thank you to the reviewer. AntiGan is a lipoprotein extract; it contains essential amino acids, natural mono- and polyunsaturated fatty acids (mainly of the omega 3 type), vitamins (B1, B2, B3 and D), and minerals (phosphorus, potassium and magnesium). AntiGan was generated by non-denaturing biotechnological processes into powder-form and presented in a capsule format for clinical use. For cell culture experiments, however, AntiGan was insoluble across a variety of solvents. We therefore sonicated AntiGan in sterile filtered 0.9% NaCl, on ice, then centrifuged this extract at 300 × g for 3 min and collected the supernatant, which we used for cell culture experiments. The high concentrations used therefore reflect the use of supernatant extract for cell culture treatments. This information has been added to the Materials and Methods section on page 3 of the revised manuscript (lines 146–148).

Comment 6:

The authors do not show the chemical composition of the evaluated extract. This information needs to be shown in the manuscript.

Response:

We appreciate this comment by the reviewer. This information has been added to the Materials and Methods section on page 3 of the revised manuscript (lines 144–146), together with a new Table 1 (page 4).

Reviewer 3 Report

Dear Editor,

The manuscript “AntiGan: an epinutraceutical bioproduct with anti-tumor properties” represents an interesting study on natural bioactive compound, AntiGan. The overall idea and design of the study are very good. However, the tests performed have some drawbacks and the data are not very convincing. The most important items that need careful revision are listed below:

  1. The title must be optimized as the two types of cancer cell lines have been studied here only. On the other hand, the main focus and novel finding in this research is limited to hepatocellular carcinoma, as indicated in lines 108-111.
  2. In introduction, regarding to GLOBOCAN 2021 (the ref. No. 3), providing global statistics on colon and liver cancer is more meaningful and better than the statistics of just one country.
  3. As the HepG2 and HCT116 cells have been taken from other labs, the cell lines should be authenticated using short-terminal DNA repeat assays.
  4. What was the source of AntiGan? Was it developed/produced in the lab or purchased from companies? In any cases the required details must be added.
  5. Table 1 does not include Sirt2 and DNMT1
  6. Some quantity has been omitted from line 204 correlated to Fig. 1B.
  7. Figure 1 contains several omissions in y axes or chart titles.
  8. In part 2.4 cell viability tests, how many times each of the experiments were repeated independently? Were the data collected from only one time running the experiment with 8 replicates of samples?
  9. In figure 1A it seems by increased concentrations of AntiGan, after 24 HepG2 cell viability was often increased. How it is explained? No statistics have been provided for 24 h experiments.
  10. The pictures of the immunostained cells Figures 2A, 3A, 4G-H, 5 against DNMT3a (red) and 6G are too dark.
  11. In line 157, a brief explain on how DNA was measured with microplate spectrophotometer or a reference must be included.
  12. Line 230; please verify the cell lines. They are not all colorectal adenocarcinoma cell lines! For example, HL60 is a blood cell line.
  13. The best way to check RNA quality, its integrity, is gel electrophoresis. While spectrophotometry and 260/280 and 260/230 ratios evaluate its purity. Here the authors have skipped the standard quality measurements of RNA. Then the reduced level of DNMT1, 3a is under question. Also the statistics of the experiments have not been reported.
  14. In immunostaining pf DNMT1, 3a experiments, the pictures must include reference bar or magnification. For subcellular localization of proteins, a higher magnification and bright clear pictures are necessary.
  15. How reliable are the mRNA quantification studies? For example, “SIRT1 mRNA levels decreased from 1, to 0.0067”.
  16. How did the authors measure sirtuin activity?
  17. Discussion contains huge amount of literature review, most of them is better to be summarized and moved to introduction part. This part needs a precise revision to focus on major findings of the study.

Author Response

REVIEWER 3

Comment 1:

The title must be optimized as the two types of cancer cell lines have been studied here only. On the other hand, the main focus and novel finding in this research is limited to hepatocellular carcinoma, as indicated in lines 108-111.

Response:

Thank you to the reviewer. The title of the manuscript has been changed to “AntiGan: an epinutraceutical bioproduct with anti-tumor properties in cultured cell lines”

Comment 2:

In introduction, regarding to GLOBOCAN 2021 (the ref. No. 3), providing global statistics on colon and liver cancer is more meaningful and better than the statistics of just one country.

Response:

In the revised manuscript, we have elaborated on the incidence of colorectal and liver cancer in more detail on page 1 of the Introduction section (lines 33-44).

Comment 3:

As the HepG2 and HCT116 cells have been taken from other labs, the cell lines should be authenticated using short-terminal DNA repeat assays.

Response:

The HepG2 cell line was kindly provided to us this year by Dr. Ana Aranda (Instituto de Investigaciones Biomédicas “Alberto Sols,” Consejo Superior de Investigaciones Científicas, Universidad Autónoma de Madrid, Spain). The present study is our first research project with this HepG2 cell line. Dr. Aranda´s group has several publications with this same cell line, which they have periodically authenticated using the StemElite ID system (Promega, Madison, WI, USA) (Martinez-Iglesias O et al, 2016).

The HCT116 cell line was a kind gift from Dr. Angelica Figueroa (Epithelial Plasticity and Metastasis Group, Instituto de Investigación Biomédica de A Coruña, A Coruña, Spain) whose group has previously published with this same cell line (Diaz-Diaz A et al, 2020). Dr. Figueroa also gave this cell line to us this year, which we then used for the first time in the current study.

Given the limited (10-day) timeframe for manuscript resubmission, it is not possible for us to perform short tandem DNA repeat profiling since we do not currently have the appropriate probes and materials in our laboratory and must, furthermore, procure them.

References

   Martínez-Iglesias O.A., Alonso-Merino E., Gómez-Rey S., Velasco-Martín J.P., Martín Orozco R., Luengo E., García Martín R., Ibáñez de Cáceres I., Fernández A.F., Fraga M.F., et al. Autoregulatory loop of nuclear corepressor 1 expression controls invasion, tumor growth, and metastasis. Proc. Natl. Acad. Sci. USA. 2016;113:E328–E337.

   Díaz-Díaz A., Roca-Lema D., Casas-Pais A., Romay G., Colombo G., Concha Á., Graña B., Figueroa A. Heat Shock Protein 90 Chaperone Regulates the E3 Ubiquitin-Ligase Hakai Protein Stability. Cancers. 2020;12:215.

Comment 4:

What was the source of AntiGan? Was it developed/produced in the lab or purchased from companies? In any cases the required details must be added.

Response:

Thank you to the reviewer. In the 2000’s, our group conducted a marine prospection program in search of nutraceuticals for the prevention and treatment of diseases. The aim was to exploit the biological properties of marine species to develop a nutraceutical that maintains all the healthy properties of the original species. Different fish and mollusk species were investigated using non-denaturing biotechnological procedures and screening protocols. A novel category of nutraceutical compounds was characterized and represented by the lipofishins (LFs), a series of complex lipoproteins derived from marine resources. The manufacturing arm of our group extracted one of these compounds (AntiGan) from the epidermis and muscular structures of the sea eel Conger conger.

We have provided the required details in the Materials and Methods section on page 3 of the revised manuscript (lines 143-146). We have furthermore included a new Table 1 (page 4) that shows the composition of AntiGan.

Comment 5:

Table 1 does not include Sirt2 and DNMT1

Response:

Thank you to the reviewer. We did not list SIRT2 or DNMT1 antibodies in Table 2 (previously Table 1) because we did not use them for immunocytochemistry.

Comment 6:

Some quantity has been omitted from line 204 correlated to Fig. 1B.

Response:

The authors apologize for the error and thank the reviewer for pointing this out. This has been corrected to “10 µg/mL” on page 6 (line 250) of the revised manuscript.

Comment 7:

Figure 1 contains several omissions in y axes or chart titles.

Response:

We apologize for the oversight. Figure 1 has been corrected and includes titles for the Y- (cell viability, %) and X-axes (AntiGan, µg/mL).

Comment 8:

In part 2.4 cell viability tests, how many times each of the experiments were repeated independently? Were the data collected from only one time running the experiment with 8 replicates of samples?

Response:

The data were collected from two independent experiments with eight replicates per experiment. This information has been added to “section 2.4 Cell viability assay” on page 5 of the revised manuscript (lines 182–183).

Comment 9:

In figure 1A it seems by increased concentrations of AntiGan, after 24 HepG2 cell viability was often increased. How it is explained? No statistics have been provided for 24 h experiments.

Response:

We have added the required statistics to Fig 1A. In HepG2 cells, we observed a slight increase in cell viability at the 24 h timepoint. In the presence of 50 µg/mL AntiGan, cell viability significantly increased by 20% (p < 0.01), and with 500 µg/mL AntiGan, cell viability significantly increased by 30% (p < 0.01). However, we did not observe any effect of AntiGan on HCT116 cell viability at the 24 h timepoint. Increase in cell viability in HepG2 cells may, speculatively, be caused by growth factors or proliferation-promoting components in AntiGan. Nevertheless, at 48 h and 72 h after treatment with AntGan, the inhibition of cell viability was evident and significant in both HepG2 and HCT116 cell lines.

Comment 10:

The pictures of the immunostained cells Figures 2A, 3A, 4G-H, 5 against DNMT3a (red) and 6G are too dark.

Response:

In the revised manuscript, we have updated each ICC image set: the brightness has been intensified and the resolution increased from 300 points per pixel to 600 points per pixel.

Comment 11:

In line 157, a brief explain on how DNA was measured with microplate spectrophotometer or a reference must be included.

Response:

We have included an explanation on how DNA was measured in this study. This information has been added to section “2.5. DNA extraction” on page 5 of the revised manuscript (lines 189–192).

Comment 12:

Line 230; please verify the cell lines. They are not all colorectal adenocarcinoma cell lines! For example, HL60 is a blood cell line.

Response:

Thank you to the reviewer. We have changed “AntiGan displays cytotoxic and apoptotic activity in colorectal adenocarcinoma cell culture lines (HL60, HS 274.T, HS 313.T, H2126, WM 115, HS 281T, Caco-2, HT-29 and SW-480)” to “AntiGan displays cytotoxic and apoptotic activity in various human cell lines including promyelocytic human leukemia (HL60), breast cancer (Hs 274.T), lung adenocarcinoma (H2126), melanoma (WM 115) and colorectal cancer (Caco-2, HT-29, SW-480).” (Section “3.2. AntiGan treatment induces apoptosis in human cancer cell lines” on page 7 of the revised manuscript; lines 275–278).

Comment 13:

The best way to check RNA quality, its integrity, is gel electrophoresis. While spectrophotometry and 260/280 and 260/230 ratios evaluate its purity. Here the authors have skipped the standard quality measurements of RNA. Then the reduced level of DNMT1, 3a is under question. Also the statistics of the experiments have not been reported.

Response:

We appreciate this helpful comment. Unfortunately, we do not have enough samples to perform gel electrophoresis to determine RNA integrity. In this study, we did however, use GAPDH as a housekeeping gene and obtained similar Ct values across different treatment conditions. Our GAPDH Ct values would be greater if there were issues related to RNA quality. We have provided a table below showing the average GAPDH values obtained for each treatment condition.

Average GAPDH values

CONDITION

GAPDH Ct

HepG2 Control

23.93

HepG2 10

23.74

HepG2 50

23.95

HCT 116 Control

25.82

HCT116 10

25.7

HCT116 50

25.12

Comment 14:

In immunostaining pf DNMT1, 3a experiments, the pictures must include reference bar or magnification. For subcellular localization of proteins, a higher magnification and bright clear pictures are necessary.

Response:

Thank you to the reviewer. We have provided new photomicrographs for each Figure. For each updated figure, the brightness has been intensified and the resolution increased from 300 points per pixel to 600 points per pixel. These images were taken under higher magnification, and scale bars have been added to the first image in an image-set; this scale bar applies to all images for that Figure.

Comment 15:

How reliable are the mRNA quantification studies? For example, “SIRT1 mRNA levels decreased from 1, to 0.0067”.

Response:

We repeated our qPCR experiments after obtaining strong inhibition of expression in response to AntiGan treatment (DNMT1 in HepG2 and HCT116 cells, DNMT3a in HepG2 cells, SIRT1 in HepG2 and HCT116 cells, and SIRT2 in HepG2 cells). Our findings were comparable, indicating that the mRNA quantification data are reliable.

Comment 16:

How did the authors measure sirtuin activity?

Response:

We have incorporated and elaborated on the measurement of sirtuin activity in a new section (2.9 Sirtuin activity assay) in the Materials and Methods section on page 6 of the revised manuscript (lines 221–235).

Comment 17:

Discussion contains huge amount of literature review, most of them is better to be summarized and moved to introduction part. This part needs a precise revision to focus on major findings of the study.

Response:

Thank you to the reviewer. We have revised the Discussion section in the submitted manuscript by summarizing and moving some of the background information to the Introduction section. In addition, in the Discussion section, we critically interpret the findings of our study in the context of published literature. We have further included a schematic working model of how AntiGan produces its broad cellular effects against cultured liver and cancer cells.

Round 2

Reviewer 1 Report

The authors have addressed many  comments 

Reviewer 3 Report

Dear Editor,

The manuscript has been well revised and is now acceptable.